# Establishment of a Halophilic Bloom in a Sterile and Isolated Hypersaline Mesocosm

**DOI:** 10.3390/microorganisms11122886

**Published:** 2023-11-29

**Authors:** Matthew E. Rhodes, Allyson D. Pace, Menny M. Benjamin, Heather Ghent, Katherine S. Dawson

**Affiliations:** 1Department of Biology, College of Charleston, Charleston, SC 29424, USA; apace21@rvc.ac.uk (A.D.P.); ghentht@g.cofc.edu (H.G.); 2Department of Drug Discovery and Biomedical Sciences, Medical University of South Carolina, Charleston, SC 29425, USA; benjamim@musc.edu; 3Institute of Earth, Ocean, and Atmospheric Science, Rutgers University, Piscataway, NJ 08854, USA; kat.dawson@rutgers.edu

**Keywords:** haloarchaea, aerobiome, Great Salt Lake, dispersal, *Haloarcula*

## Abstract

Extreme environments, including hypersaline pools, often serve as biogeographical islands. Putative colonizers would need to survive transport across potentially vast distances of inhospitable terrain. Hyperhalophiles, in particular, are often highly sensitive to osmotic pressure. Here, we assessed whether hyperhalophiles are capable of rapidly colonizing an isolated and sterile hypersaline pool and the order of succession of the ensuing colonizers. A sterile and isolated 1 m^3^ hypersaline mesocosm pool was constructed on a rooftop in Charleston, SC. Within months, numerous halophilic lineages successfully navigated the 20 m elevation and the greater than 1 km distance from the ocean shore, and a vibrant halophilic community was established. All told, in a nine-month period, greater than a dozen halophilic genera colonized the pool. The first to arrive were members of the Haloarchaeal genus *Haloarcula*. Like a weed, the *Haloarcula* rapidly colonized and dominated the mesocosm community but were later supplanted by other hyperhalophilic genera. As a possible source of long-distance inoculum, both aerosol and water column samples were obtained from the Great Salt Lake and its immediate vicinity. Members of the same genus, *Haloarcula*, were preferentially enriched in the aerosol sample relative to the water column samples. Therefore, it appears that a diverse array of hyperhalophiles are capable of surviving aeolian long-distance transport and that some lineages, in particular, have possibly adapted to that strategy.

## 1. Introduction

Hypersaline environments are characterized by having a salinity above that of ocean water or greater than 38 ppt [1]. From there, they range up to the most saline bodies of water on the planet, such as Don Juan Pond in the dry valleys of Antarctica and Gaet’ale Pond in the Danakil Depression of Ethiopia, with salinities topping 400 ppt, more than twelve times more saline than average ocean water [2,3]. Hypersaline bodies of water may be both relatively large (i.e., the Dead Sea is 600 km^2^ and the Great Salt Lake is 4400 km^2^) and enduring (the Dead Sea has existed in its current state for several million years, and the Great Salt Lake has remained hypersaline for roughly 10,000 years) [4,5]. At the same time, small, transient, and hypersaline environments are constantly forming. Any time a tidepool or salt marsh is cut off from the ocean and appreciably evaporates, a hypersaline environment is temporarily formed. 

Once a greater than 2 M concentration of sodium chloride is achieved, these environments can become host to vibrant communities of hyperhalophilic life not fully appreciated until the advent of culture-independent methods and further elucidated by more recent metagenomic approaches [6,7]. Extreme halophiles require the presence of at least 1.5 M NaCl and can grow at salinities up to and even slightly beyond the saturation point for NaCl (5.5 M) [8]. Under optimal conditions and at moderate salinities (2–4 M NaCl), microbial cell densities can surpass 10^7^/mL and are generally dominated by diverse members of the archaeal class Halobacteria [9,10]. At high cellular densities, the hypersaline bodies of water often become tinged pink due to the carotenoid containing rhodopsin proteins in the cell walls of the Haloarchaea [11]. 

Hypersaline environments themselves are chemically diverse and are not exclusively comprised of NaCl-based brines. Brine solutions are often rich in magnesium (Mg^2+^), sulfate (SO_4_^2−^), and potassium (K^+^) ions, and the ionic mixture is often pivotal in determining the makeup of the halophilic microbial community [12]. The Dead Sea, for example, is composed of approximately 53% magnesium chloride, 37% potassium chloride, and 8% sodium chloride [13]. While most hypersaline environments on Earth are, in fact, dominated by NaCl, the diversity of habitable brine environments expands both the habitable range here on Earth and the likelihood of microbial life existing in extraterrestrial environments [12,14].

As with other extreme environments, such as frigid mountaintops or arid deserts, hypersaline bodies of water are often biogeographically isolated. Hundreds or even thousands of kilometers of relatively inhospitable (from an extremophile perspective) terrain may surround a hypersaline lake. Nevertheless, nearly identical strains of hyperhalophiles have been isolated from globally distributed regions [15].

This then begs the interrelated ecological questions of (1) how are hyperhalophiles transported and introduced globally? (2) How can they possibly survive the osmotic stress of transport through moderate, less saline environments? And (3) what dynamics impact the colonization and development of hypersaline microbial communities?

In the past and most recently in the years 1980 and 1992, *Dunaliella* blooms occurred in the Dead Sea due to the dilution of the Dead Sea surface waters by freshwater and an influx of inorganic phosphate from environmental runoff. Satellite imagery of a developing Dead Sea bloom of the hyperhalophilic green microalgae *Dunaliella* in 1991–1992 indicated that the algal bloom was initiated in the shallow sediments surrounding the lake [16]. The algal bloom was rapidly followed by a bloom of the heterotrophic Haloarchaea, which feed upon the osmolyte glycerol that is produced by the *Dunaliella* [17]. During the 1991–1992 bloom, the Dead Sea bloom was reported to contain up to 3 × 10^7^ red archaea cells/mL [9].

The normal ion content and pH of the Dead Sea, which, while saturated for NaCl, contains an atypical ionic composition with higher concentrations of the divalent cations (Mg^2+^ and Ca^2+^) relative to monovalent cations (Na^+^ and K^+^) and slightly acidic water (~6.0), prevents *Dunaliella* and other halophilic microorganisms from regularly blooming. More recently, changes in rainfall patterns and water usage have prevented blooms from developing in the Dead Sea for the past three decades.

These observations suggested that there remained dormant repositories of halophilic cells in adjacent environments even as the salinity and chaotropic nature of the Dead Sea increased to a level that was inhospitable to the most resolute of hyperhalophiles [18]. There exist a number of surrounding and underground freshwater springs that feed into the Dead Sea, providing less saline and less challenging havens for hyperhalophilic life even as the remainder of the Dead Sea becomes largely inhospitable [19]. There may remain other dormant or transient hyperhalophilic communities spread throughout the globe. 

With regard to mechanisms of long-distance transport, oceanic transport may, in fact, be possible for some strains of hyperhalophiles. Members of the genus *Halococcus* can survive immersion in fresh water for extended periods of time, and viable hyperhalophilic cells of a variety of strains have been isolated from sea-water [20,21,22,23]. However, other lineages of hyperhalophiles readily lie below 100 ppt salinity, rendering oceanic transport of these strains unlikely. These include strains of some of the most ubiquitous members of the Haloarchaea, such as members of the genera *Haloquadratum*, *Haloarcula*, and *Halorubrum*, among others [15,24,25,26,27].

Potentially viable Haloarchaea have been isolated from both human fecal matter and the human intestinal mucosa [28,29,30]. Irrespective of whether the human-associated Haloarchaeal members are transiting members ingested by humans during salt consumption or have taken up permanent residence in the human gut, animal salt ingestion may provide a method of long-distance dispersal. Similarly, Kemp et al. suggested that migratory birds may trans-continentally transport hyperhalophiles [31]. Migratory birds often land in hypersaline lakes like the Great Salt Lake, and viable hyperhalophiles, thus far exclusively from the genus *Haloarcula*, have been cultivated from salt crystals adhered to the feathers of Great Salt Lake White Pelicans [31]. In addition, halophiles of the genus *Halococcus* were reportedly isolated from the nostrils of a migratory seabird, *Calonectris diomedea*. Salt-excreting glands may serve as a hypersaline environment for microbes to travel between distant habitats previously separated via allopatric speciation [32].

The Haloarchaea may, in fact, be ideally suited to dispersal via Earth’s stratosphere. Not only have members of the Haloarchaea adapted to life in hypersaline conditions, but they are also often multiple extremophiles. Many members of the Haloarchaea are capable of withstanding additional adverse conditions such as desiccation and radiation exposure [33,34,35,36]. Others, such as *Halohasta litchfieldiae*, are also psychrophilic [37]. Cold-adapted Haloarchaea survived largely intact after prolonged exposure to Earth’s stratosphere, and the viability of mesophilic Haloarchaea was only slightly impacted [38]. Moister conditions in the troposphere may at times hinder the aeolian dispersal of low osmolarity sensitive hyperhalophiles, but as with other categories of microorganisms [39], aeolian dispersal most likely figures in the long-distance transport of halophiles.

In turn, the primary long-distance dispersal mechanisms of the hyperhalophiles should figure prominently in the establishment of novel hypersaline communities [40]. Hyperhalophiles that are more easily dispersed and can survive transport will be the first to arrive at newly generated hypersaline pools and potentially gain an early competitive advantage. In order to assess the order of succession and the potential for transport of hyperhalophilic microorganisms, we constructed a 1 m^3^ sterile (with regard to halophiles) hypersaline pool about 50 feet above ground level, atop the Saint Philip’s Street Parking Garage in Charleston, South Carolina. The pool was regularly monitored both via microscopy and molecularly, and within months, a dense population of Haloarchaea had established residence.

As the Great Salt Lake represents by far the largest repository of halophilic life in the North American continent, it may well serve as a regular source of inoculum for the long-distance dispersal of halophiles. The mesocosm pool, located in Charleston, South Carolina, lies in the prevailing downwind direction of the Great Salt Lake. Therefore, both aerosolized and water column microbial communities from the northern basin of the Great Salt Lake were obtained in the same summer. Our experiment was meant as a proof of concept, demonstrating that aerial transport of hyperhalophiles does, in fact, occur and to establish the order of succession of a pristine hypersaline environment.

## 2. Materials and Methods

### 2.1. Construction of the Hypersaline Mesocosm Pool

A 1 m^3^ plastic tub was placed in the northwest corner atop the Saint Philip’s Street Parking Garage in Charleston, SC (32.7863° N, 79.9391° W). The tub was elevated above the tree line at about 50 feet off the ground and located approximately a mile from the nearest Atlantic Ocean shoreline (Figure 1).

During the month of March 2019, the plastic tub was cleaned with bleach and rinsed prior to the addition of approximately 800 L in 50 L increments of highly purified water. A Barnstead^TM^ MegaPure^TM^ Glass Still (Lake Balboa, CA, USA) purified the water via distillation, deionization, and reverse osmosis. Concurrently, the inorganic chemicals listed below were added to the distilled water (Table 1). The salt mix was based upon a combination of a generic haloarchaeal media (DSMZ media 1018) with the addition of a nitrate source (KNO_3_), a phosphate source (KH_2_PO_4_), a carbonate source (NaHCO_3_), and manganese (MnSO_4_) to encourage the growth of both other lineages of the Haloarchaea and the halophilic algae *Dunaliella salina.* Given the quantities required to construct such a pool de novo, chemical choices were made partially based on availability and cost. Morton^®^ Select Sea Salt (Chicago, IL, USA), in particular, is one of the few commercially available Sea Salts that are available in large quantities, are manufactured by simply evaporating sea water to dryness, and contain no additives. The assumption was that all necessary trace elements would be provided either in the Morton^®^ Select Sea Salt or by aeolian deposition. 

The higher quantity chemicals (Morton^®^ Select Sea Salt and MgCl_2_) were baked at 140 °C for a minimum of 12 h. All other chemicals were dissolved and autoclaved prior to addition to the pool. A hand-held powered cement mixer was initially utilized to aid in dissolving the solutes, and subsequently, an aquarium pump was installed to ensure constant mixing of the water and the prevention of a freshwater lens from forming on top following rainfall events.

For the first iteration of the pool, organic carbon was provided by the addition of 2 kg of yeast extract. The yeast extract was dissolved in Millipore Sigma Milli-Q^®^ (Burlington, MA, USA) water and autoclaved. A second iteration of the pool, initiated in the spring of 2021, replaced the yeast extract with a combination of 1 kg of glycerol and 1 kg of glucose. Again, both organic carbon sources were autoclaved prior to addition to the pool.

Precautions were taken to ensure that halophiles were not introduced to the mesocosm pool from cultures growing in the lab. All pool construction and sampling excursions took place at the start of the day, prior to entry into the lab and work with halophilic cultures. In addition, a shower was required after working in the lab before returning to the mesocosm pool the following day.

### 2.2. Monitoring of the Mesocosm Pool

The pool was visually and microscopically inspected biweekly for evidence of growth. The salinity and pH of the pool were measured intermittently to ensure that there were no drastic changes, using a hand-held refractometer and a Horiba LAQUAtwin-22 hand-held pH meter (Kyoto, Japan), respectively. A pool skimmer was used to remove surface debris (insects, leaves, garbage, and even one time a squirrel, etc.). Once growth was observed microscopically, samples were collected for both cell counting and molecular analysis on a regular basis.

### 2.3. Controls

To demonstrate the sterility of the pool with regard to hyperhalophiles, two liters of pool media were prepared in the laboratory following a similar protocol to the creation of the mesocosm pool. Subsequently, 100 mL aliquots were incubated in a shaker at 90 rpm at 37 °C for several months. In addition, early samples from the pool collected during the first week after the addition of an organic carbon source and prior to visual confirmation of growth were collected and incubated at 37 °C. The pool media was also tested on a variety of halophiles currently being cultured in the lab, including *Haloferax sulfurifontis* and *Halococcus moorhuae*. Once growth was established in the pool, freshly synthesized pool media was inoculated with a pool culture to demonstrate growth.

### 2.4. Great Salt Lake Sampling

As a large-scale repository of halophilic life on the North American continent located generally upwind of the Eastern United States, the Great Salt Lake represents one potential source of long-distance inoculant to the mesocosm pool and other transient hypersaline environments. Surface water samples from a depth of approximately 6 inches were collected in the north arm of the Great Salt Lake in the vicinity of the Spiral Jetty on 23 July 2019 (Figure 1d). Samples were collected from 100 m, 200 m, and 300 m from shore. For each sample, approximately 500 mL of water was filtered through a Whatman^®^ 0.2 μm polycarbonate filter (Maidstone, UK). Samples were stored and shipped on dry ice prior to long-term storage at −80 °C at the College of Charleston until processing.

### 2.5. Aerosol Sampling

Aerosol samples were collected from the Great Salt Lake on 23 July 2019. Sample collection occurred approximately 50 feet from the shoreline of the Lake, just adjacent to the location of the water sampling. Aerosol samples were collected from the Bonneville Salt Flats on 24 July 2019. Sample collection occurred in the Bonneville State Park, approximately 300 m north of the Salt Flats Rest Area. Biological aerosol sampling followed the protocol of Šantl-Temkiv et al. [41]. Samples were collected with a Karcher DS5800 impinger vacuum (Winnenden, Germany) plugged into a portable generator. Approximately 1500 L/min of air was filtered through 1.5 L of a PBS solution with a salinity of 150 g/L NaCl. At the Bonneville Salt Flats, the impinger was run for 2 h, and at the Great Salt Lake, the impinger was run for 3 h. Subsequently, an aliquot of 500 mL of the PBS solution was filtered through a Whatman^®^ 0.2 μm polycarbonate filter. Samples were stored and shipped on dry ice prior to long-term storage at −80 °C at the College of Charleston until processing.

### 2.6. Molecular Analysis

For each pool collection, 50 mL of pool sample was centrifuged, and DNA was extracted from the resulting cell pellet using a DNAeasy Power Soil Pro Kit (Qiagen, Hilden, Germany). Likewise, DNA was extracted from the Great Salt Lake and aerosol filter samples using the DNAeasy Power Soil Pro Kit. Amplification of the V4 region of the 16S rRNA gene for each sample was conducted in triplicate using dual barcoded primers according to the protocol of the Earth Microbiome Project [42]. Triplicate PCR samples were pooled, and then all mesocosm, Great Salt Lake, and aerosol samples were pooled together and sequenced at the Dartmouth Core Facility on an Illumina miniSeq (San Diego CA, USA). All downstream analyses were performed using the QIIME2 platform (version 2023.9) [43].

Sequences were quality controlled using the Deblur package, and the joined reads were truncated at 250 base pairs. This left more than 95,000 quality-controlled sequences for each mesocosm sample and more than 60,000 quality-controlled sequences for the Great Salt Lake water sample. The aerosol samples had fewer sequences, with the Great Salt Lake aerosol sample having the fewest with only approximately 4500 sequences, half of which were removed for quality control. Amplicon sequence variants (ASVs) taxonomy was assigned based on the SILVA 138v 16S rRNA database. The raw Illumina 16S rRNA barcode sequences and metadata collected in this study are available from the NCBI Small Read Archive (BioProject # PRJNA1037157).

## 3. Results

The first iteration of the hypersaline mesocosm pool at the College of Charleston (CofC), affectionately nicknamed the Dead CofC, was fully filled by the end of March 2019. The addition of yeast extract as a source of organic matter proved to be an unfortunate choice as it permanently clouded the Dead CofC, rendering it difficult to casually observe growth and imparted a brown color to the water (Figure 2a). In the second iteration of the mesocosm pool in which glycerol and glucose were used instead of yeast extract, the resulting halophilic bloom did, in fact, impart the characteristic red color of a haloarchaeal bloom to the pool (Figure 2b).

It took approximately two months for growth to be observed microscopically. Growth was confirmed via molecular analysis on day 65 (Figure 3). Initially the dead CofC was dominated by members of the genus *Haloarcula* comprising greater than 99.97% of the population (Figure 3A). Members of the genera *Halorubrum* and *Natronomonas* were also observed at that time making up a fraction of a percent of the total population. The pool remained dominated by *Haloarcula* for several months before *Haloarcula* was largely supplanted by *Halorubrum*.

Over the course of the next several months, additional halophilic lineages arrived, including halophilic members of the bacterial genera *Longimonas* and *Halomonas,* and the family Balneolaceae. Roughly a month and a half after the establishment of the bloom, cell counts began to decrease (Appendix A). The decline continued until about four months after the pool was filled when the halophilic algae *Dunaliella* first appeared. The presence of the primary producer stabilized the cell counts moving forward. All told, a total of at least 18 prokaryotic genera established populations (>0.1% at any given time) in the pool. During this time period, both the salinity and pH of the mesocosm were intermittently monitored. As anticipated, the salinity fluctuated slightly due to evaporation and rainfall (between 180 and 220 ppt). The aquarium pump prevented a freshwater lens from forming on the top of the pool. The pH remained constant at just over 7.

Media prepared identically to pool media demonstrated no growth after months of incubation. Early samples of pool media collected during the months of March and April, prior to observed growth, also demonstrated no growth after months of incubation. In contrast, *Haloferax sulfurifontis* and *Halococcus morrhuae* both readily grew in pool media and pool media inoculated with samples collected in mid-June 2019, when the pool was dominated by *Haloarcula* and demonstrated rapid growth.

Not surprisingly, all three-north arms of the Great Salt Lake samples closely resembled each other and were dominated by members of the Haloarchaea (Table 2). Members of the genus *Halonotius* were most prevalent, representing more than one-third of the population. Following *Halonotius* in prevalence was the genus *Haloquadratum,* which comprised approximately one-eighth of the population and the genera *Haloparvum, Halorubrum*, and the bacteria genus *Salinibacter*, each constituting on the order of one-twentieth of the population. *Haloarcula* were, in fact, present but at comparatively low concentrations. The Great Salt Lake Aerosol sample demonstrated a broader population diversity. Nearly half of the sample was composed of non-Haloarchaeal members, including several human-associated lineages. Thus, virtually every lineage of Haloarchaea had a smaller proportion in the aerosol sample than in the Great Salt Lake samples, except for one. Members of the genus *Haloarcula* demonstrated, on average, an approximately hundred-fold and statistically significant increase in their proportion in the aerosol sample relative to the water column samples (3.36% vs. 0.036%) (Appendix A). The Bonneville Salt Flat aerobiome sample had a highly diverse population with the Haloarchaea being virtually unrepresented (Appendix A). 

## 4. Discussion

The lack of growth in the mesocosm control media indicates that the growth in the pool was not due to contamination of the media from the Morton^®^ Select Sea Salt or other media components. Furthermore, as the strains growing in the pool were not related to any culture currently present and growing in the lab at the College of Charleston, growth was not accidentally introduced. We therefore conclude that, as intended, hyperhalophiles were deposited in the pool from the environment.

The mesocosm media was a relatively simple media lacking a variety of potentially necessary nutrients and trace elements. However, by the time the pool was finished being filled, it had been exposed to the elements for a month, and by the time cells were observed, it had been exposed to the elements for three months. This would have provided ample time for dust and other debris to accumulate a variety of trace elements. Furthermore, the fact that growth in the laboratory of isolates and early pool inoculation occurred readily and quickly suggests that it was not a nutrient limitation hindering growth but rather a dispersal effect. With that said, no environment, whether natural or artificial, can ensure the growth of all organisms. There may have been viable halophilic microbes that were deposited in the pool but were unable to grow at the time.

The first microorganisms to seemingly arrive and establish residence were members of the genus *Haloarcula*. Viable *Haloarcula* were present on the wings of the migratory birds that had landed in the Great Salt Lake, and *Haloarcula* was the only member of the Haloarchaea that was highly enriched in the Great Salt Lake aerosol sample relative to the water column [31]. Furthermore, in the early stages of the mesocosm, the *Haloarcula* dominated the ecosystem, representing greater than 99% of the haloarchaeal community for several weeks and greater than 50% of the haloarchaeal community for several months.

However, once other later arriving hyperhalophiles established residence, the *Haloarcula* population steadily decreased. Ultimately, *Haloarcula* was largely supplanted by *Halorubrum* in an inverse relationship of sequence abundances, as portrayed in Figure 3. Interestingly, similar inverse seasonal taxonomic relationships have been observed in the hypersaline Lake Tyrrell, located in Australia, between *Haloquadratum* and *Halorubrum*-related strains [44].

Once established, the *Haloarcula* remained present throughout the duration of the experiment. Towards the later observation period, as with the samples collected from the Great Salt Lake, the *Haloarcula* represented a fraction of a percent of the population. These results suggest that *Haloarcula* may be a hyperhalophile particularly adapted to dispersal. Its colonization strategy appears to be that of a weed, to arrive first and rapidly colonize a hypersaline body of water before being supplanted by other hyperhalophiles.

Nevertheless, it still took on the order of months for *Haloarcula* to establish a significant population in the pool. Was this because it took time to arrive or because conditions were not optimal for the growth of *Haloarcula* for a couple of months? The fact that early pool growth grew rapidly in identical media in the laboratory suggests the former, but as conditions in the lab versus the mesocosm were not identical, the latter possibility cannot be ruled out.

What about the *Haloarcula* seemingly lends itself to aerosolization and dispersal? Perhaps the *Haloarcula* preferentially reside closer to the water surface than other halophiles, facilitating their transfer to feathers and their aerosolization. A fine-scale depth analysis of hypersaline ecosystems could yield insight into this matter. Perhaps it has something to do with the cellular wall structure or shape of the *Haloarcula* that lends itself to initiating transport or it could be due to the viability of the *Haloarcula* and that they are more readily able to survive transport.

The near total absence of halophilic archaea from the Bonneville Salt Flats, only ~125 miles from the Great Salt Lake sampling location at Spiral Jetty, demonstrates how infrequent long-distance transport of viable halophiles may be. Nevertheless, it only takes the arrival of one viable cell to colonize a new environment.

The mesocosm pool, at a 50-foot elevation, resided ~1 mile from the Atlantic Ocean. We, therefore, cannot rule out the deposition of viable halophiles from sea spray. However, a number of the genera that were observed in the mesocosm pool seem unlikely to be able to survive prolonged exposure to seawater, suggesting that many, if not most, of the arriving halophiles would have been deposited via aeolian means. Perhaps dotted throughout South Carolina’s extensive network of salt marshes are repositories of viable hyperhalophiles. Perhaps the colonizers arrived from further afield. While samples from the Great Salt Lake contained similar taxa, our analysis identified distinct and non-overlapping ASVs between the two sites. However, we cannot fully rule out anthropogenic transport from another hypersaline environment or other deposition via larger debris (e.g., insects, small mammals, leaf deposition).

Our intention was that this experiment would provide proof of concept that sterile and isolated hypersaline bodies of water would become colonized by a diverse assortment of halophiles in a timely fashion. Ultimately, the cost for one iteration of the pool was on the order of $500, plastic tub included, and the pink waters of a haloarchaeal bloom makes for a visually appealing experiment (assuming the use of glycerol and glucose rather than yeast extract). With scale, that cost would be anticipated to go down. A network of such pools spread throughout a region or a continent would serve both to shed light on the dispersal mechanisms of halophiles and any potential patterns of dispersal.

## Figures and Tables

**Figure 1 microorganisms-11-02886-f001:**
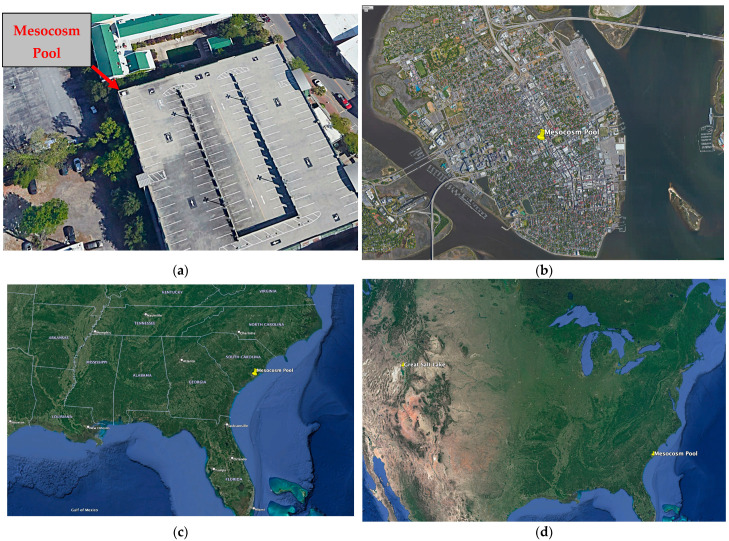
(**a**) Aerial view of the Saint Philip’s Street Parking Garage in Charleston, SC. The mesocosm pool is in the northwest corner; (**b**) Location of the mesocosm pool on the Charleston, SC peninsula showing distance from the Ashley and Cooper Rivers; (**c**) Regional scale location of the mesocosm pool in the Southeastern United States; (**d**) Location of the mesocosm pool and the Great Salt Lake on the North American continent.

**Figure 2 microorganisms-11-02886-f002:**
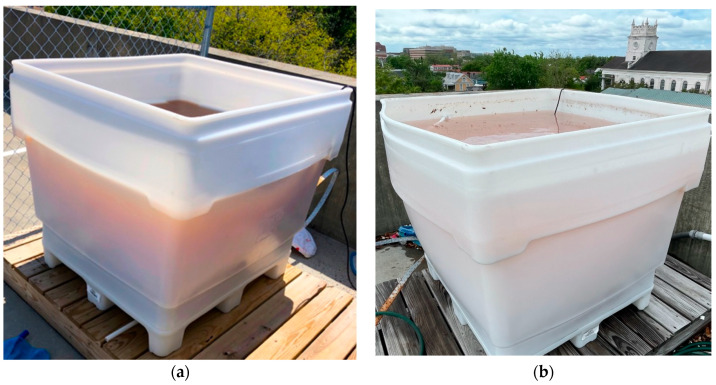
(**a**) Hypersaline mesocosm pool from first experimental run. The brown color of the water was imparted by the 2 kg of yeast extract that was added as an organic source. (**b**) Hypersaline mesocosm pool from the second experimental run. With the addition of the uncolored organic sources glycerol and glucose, the water obtained the characteristic red color of a haloarchaeal bloom.

**Figure 3 microorganisms-11-02886-f003:**
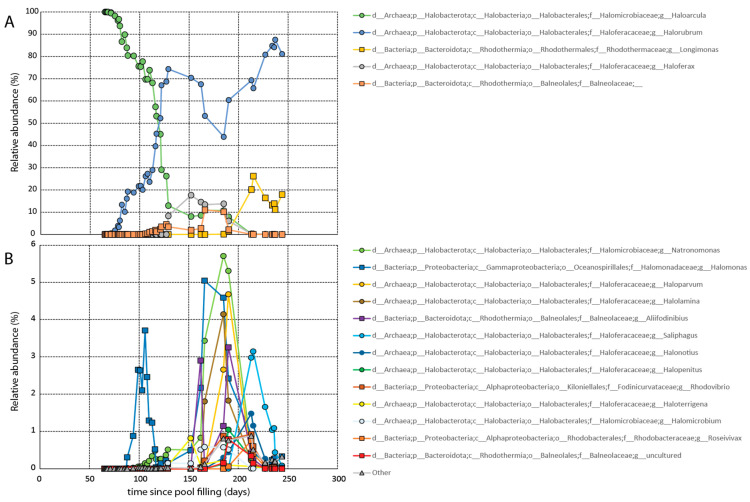
Relative abundance over time of archaeal and bacterial taxa in the mesocosm pool. (**A**) Taxa that achieved relatively large populations (>10%) at any point during the experiment. (**B**) Halophilic bacterial and archaeal taxa that displayed measurable but smaller populations during the experiment.

**Table 1 microorganisms-11-02886-t001:** List of mesocosm media components added to 800 L of distilled H_2_O.

Component	Quantity
Morton^®^ Select Sea Salt	160 kg
MgCl_2_	12 kg
MgSO_4_	400 g
CaCl_2_	80 g
KCl	1.6 kg
KNO_3_	800 g
NaHCO_3_	34.4 g
KH_2_PO_4_	28 g
MnSO_4_	1.6 g

**Table 2 microorganisms-11-02886-t002:** Percent composition of each genus and total assigned reads in the three water column samples collected 100 m, 200 m, and 300 m from shore and the aerosol sample.

	100 m	200 m	300 m	GSL Aerosol
*Haloarcula*	0.08	0.01	0.02	3.36
*Halobellus*	3.55	3.39	3.20	2.26
*Halonotius*	40.09	38.52	33.27	19.44
*Haloquadratum*	14.90	12.08	21.28	6.48
*Haloparvum*	5.63	6.22	6.06	2.58
*Halorubrum*	5.64	4.96	4.19	5.00
*Natronomonas*	3.34	2.28	2.12	1.72
*Salinibacter*	2.09	10.33	6.78	4.14
*Haloplanus*	0.51	0.34	0.36	0.11
Other Haloarchaea	23.41	19.88	20.94	14.19
Other	2.85	12.32	8.56	44.96
Total Assigned Reads	49,659	68,240	87,680	1281

## Data Availability

All sequences will be uploaded to the NCBI SRA upon acceptance of the manuscript.

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
