# Peer review of "Establishment of a Halophilic Bloom in a Sterile and Isolated Hypersaline Mesocosm"

_microorganisms, 2023, doi:10.3390/microorganisms11122886_

Round 1

Reviewer 1 Report (Previous Reviewer 2)

Comments and Suggestions for Authors

Dear authors, dear Editor,

Below, I present my evaluation of the work “Establishment of a halophilic bloom in a sterile and isolated hypersaline mesocosm” by Rhodes et al.

In the previous version it was difficult to follow the results due to their presentation and a more in-depth review of the literature was needed.

The authors improved the manuscript substantially and now it is clearer, not only the objective but also the method, the presentation of results and the scope of the document.

There are still some errors throughout the writing, but they can be improved with careful revision, for example, checking that ml or mL is written throughout the document.

I consider this article to be important for the community interested in the study of halophiles; particularly for those seeking to understand how the establishment and turnover of species takes place over time.

Once again, I thank you for the opportunity to review the manuscript and comment on possible improvements to it.

Kind regards.

Comments on the Quality of English Language

The article is well written, just need minor corrections.

Author Response

Thank you for your constructive reviews in helping to improve this manuscript. We have fixed the unit errors and inconsistencies (ml ==> mL) etc. and other grammatical errors. 

Reviewer 2 Report (New Reviewer)

Comments and Suggestions for Authors

The manuscript titled "Establishment of a halophilic bloom in a sterile and isolated hypersaline mesocosm" presents an intriguing experiment that investigates the colonization dynamics of hyperhalophiles in a controlled hypersaline pool. The study is well-structured and provides valuable insights into the succession of colonizers, their potential for long-distance transport, and adaptation strategies. 

Minor comments:

  1. The manuscript incorrectly refers to the class as Haloarchea. The correct name is Halobacteria. (https://lpsn.dsmz.de/class/halobacteria). Additionally, in line 280, the correct species name is Halococcus morrhuae.

  2. The legend for Figure 3 could be improved by making it shorter and more concise. 

  3. It is interesting to know whether the study inspected surface debris (line 186: insects, leaves, garbage, and one time a squirrel), for the presence of hyperhalophiles.

  4. Do the aerosolization and dispersal of Haloarcula exhibit any seasonal dependence?

Author Response

Thank you for your insights and corrections. We have responded them both in the manuscript and here. The questions raised in items 3 and 4 are the subject of ongoing and/or potential future directions.

1) Those two oversights/typos have been corrected.

2) The figure caption has been updated to read:

Figure 3. Relative abundance over time of archaeal and bacterial taxa in the mesocosm pool. (A) Taxa that achieved relatively large populations (>10%) at any point during the experiment. (B) Halophilic bacterial and archaeal taxa that displayed measurable but smaller populations during the experiment.

3) We wondered that ourselves. By the time debris was noticed in the pool it would have been covered in halophiles. We considered installing some kind of screen but ultimately decided against it as larger debris could represent a form of transport (the bird feathers for example). We have added a line about this to the discussion (364-365).

4) That is a great question and one in which we hope to address in multiple approaches in the future. 

Reviewer 3 Report (New Reviewer)

Comments and Suggestions for Authors

The manuscript assessed whether hyperhalophiles are capable of rapidly colonizing an isolated and sterile hypersaline pool and the order of succession of the ensuing colonizers. The study is well described and clearly supported in all its aspects, exhaustively describing the procedure performed. Unfortunately works like this, where the most advanced technology is used with relatively high analysis costs that do not allow a large number of samples, it must be based only on a small number of these; the data would have been different if supported by a higher sample base; unfortunately it is still the limit of these studies. Despite everything, the results obtained are extremely important and support the conclusions that the authors provide.

There are, however, some concerns about the quality of the manuscript.

1. Spaces are required between numbers and units, please keep the entire text consistent.

2. ml” (Line 47and “mL” (Line 77) need to be consistent.  Please review the entire text and make corrections.

3. Figure” (Lines 144, 244, 247and “Fig” (Lines 255, 257) need to be consistent.  Please review the entire text and make corrections.

4.  Please use a standardized three line table for Table 2.

5. Some references require abbreviations for journal namessuch as Ref. 5, 16, 28 and 41.

Comments on the Quality of English Language

Minor editing of English language required.

Author Response

Thank you for your insights and corrections. We view this study as a proof of concept and now that it has proved successful we would like to pursue funding for multiple pool locations and repeat aerosol collections.

1and2) We have corrected the units throughout the manuscript.

3) We have changed all references to figures that were not directly mentioned  in the text to Fig.

4) Table 2's format has been updated. 

5) References have been edited and updated. 

This manuscript is a resubmission of an earlier submission. The following is a list of the peer review reports and author responses from that submission.

Round 1

Reviewer 1 Report

Comments and Suggestions for Authors

Dear Editor,

In this work, Rhodes and co-authors described the ability of hyperhalophiles to survive at long-distance aeolian transport. This could be an evolutionary strategy adopted by some of these organisms.

Abstract: the aim of the work is elusive.

Introduction: I’ve noticed that there are several paragraphs without references. In addition, several references are old, I understand that they are milestones, but I wonder if there are any more recent references. 

Paragraph 2.1: This paragraph appears to be cryptic. I suggest describing first the salt composition and after that the carbon source used. Glycerol and glucose concentration should be reported. What about the nitrogen source?

Line 156: How did the Authors measure pH and salinity? What about the temperature?

Line 161: What the Authors mean with “was synthesized”? 

Figure 3: these results should be better represented, such as with a timeline. In addition, several species shown in the legend appear to be missing from the calendar. For example, Natronomonas in March, Halanaerobium in July and Haloferax in August. In August, what does the gray bar represent?

Line 247: Please explain why samples from the Great Salt Lake were analyzed. Do the authors hypothesize that the species in the pond are derived from this lake? If this is the hypothesis please indicate this lake on the map in Figure 1.

Discussion: The authors should discuss what are the weaknesses and strengths of this approach.

Reviewer 2 Report

Comments and Suggestions for Authors

Dear authors and editor,

Thank you for the invitation to review the manuscript entitled “Establishment of a halophilic bloom in a sterile and isolated 2 hypersaline mesocosm” by Rhodes et al. The authors present the results obtained by studying the ecological succession of perhalophilic microorganisms in an artificial system. To do this, they designed a pool with a hypersaline composition and regularly monitored the composition of microorganisms through microscopy and molecular studies. The study is important and novel, however, several points must be addressed before being published.

Next, I list a series of points that must be clarified.

1. The authors describe the location of the studied system, but it is necessary to add a map of the area, for those not familiar with the territory of the United States of America.

2. Although some key articles are presented in the Introduction, a more extensive review of the literature is required, not only for the conceptual framework, but to increase the discussion and its scope.

Benlloch, S., Martínez-Murcia, A.J., & Rodríguez-Valera, F. (1995). Sequencing of bacterial and archaeal 16S rRNA genes directly amplified from a hypersaline environment. Systematic and applied microbiology, 18(4), 574-581.

DasSarma, S., & Arora, P. (2001). Halophiles. e LS.

Foster, I. S., King, P. L., Hyde, B. C., & Southam, G. (2010). Characterization of halophiles in natural MgSO4 salts and laboratory enrichment samples: astrobiological implications for Mars. Planetary and Space Science58(4), 599-615.

Oren, A. (2014). The ecology of Dunaliella in high-salt environments. Journal of Biological Research-Thessaloniki21(1), 1-8.

Podell, S., Emerson, J. B., Jones, C. M., Ugalde, J. A., Welch, S., Heidelberg, K. B., ... & Allen, E. E. (2014). Seasonal fluctuations in ionic concentrations drive microbial succession in a hypersaline lake community. The ISME journal8(5), 979-990.

Ventosa, A., de la Haba, R. R., Sanchez-Porro, C., & Papke, R. T. (2015). Microbial diversity of hypersaline environments: a metagenomic approach. Current Opinion in Microbiology25, 80-87.

3. To perform the experiments, the authors made modifications to the generic haloarchaeal media (DSMZ media 1018, https://www.dsmz.de/microorganisms/medium/pdf/DSMZ_Medium1018.pdf), it would be interesting if they explained if the changes occurred due to previous experiences, please explain what their choices are based on.

4. In the methods section (l-142-143) authors state: “All chemicals were either baked at 140C for a minimum of 12 hours or dissolved and 142 autoclaved prior to addition to the pool, please explain which reagents were baked and which ones were autoclaved. Information is missing from some points to be able to replicate the experiment.

5. The authors comment that a second experiment was carried out, in which glycerol and glucose were added, instead of the yeast extract, but they do not say what amounts were added.

6. Controls section (L-161-162). Please explain the amount (volume) that you prepared and the conditions of incubation.

7. Results. In Fig. 3 there is a list of the different organisms that are detected, but not all of them are represented. For example, in the month of June Natronomonas is mentioned (in purple) but it is not on the calendar. Review the information.

8. Fig. 4 is not clear, a number of taxa are shown, but apparently only Haloarculax and Natronomonas are recorded, the others are listed but not seen on the graph. Please look for a clearer way to represent the data.

9. Initially I did not understand Table 2, after re-reading the text I realized that the notations 100 M, 200 M and 300 M refer to water samples collected at different depths. Please use the international system of units, to avoid confusion, in the SI M=molar and m=meters. On the other hand, this table would be more complete if the count of microorganisms for the artificial pool were added, to the counts in natural environments.

10. Discussion. The authors note that there was no growth of microorganisms in the control system (the Haloarcula dominated the ecosystem representing greater than 99% of the haloarchaeal community for several weeks and greater than 50% of the haloarchaeal community for several months), but do not present the data.

11. In the discussion the authors indicate a time in which the organisms remain in the pool, but they do not show a table or figure for it, so it is not easy to follow their arguments.

12. The discussion should take advantage of the authors' data and compare it with other experiments, even if these have not been carried out in the same area.

13. I did not have access to the supplementary material, so some of my doubts could be answered if this material was available.

14. The figures are not clear or self-explanatory; it is not easy to understand the content, since there is  omission in the representation of contents and in the figure captions.

Comments on the Quality of English Language

The article is well written, in terms of handling English. But the nomenclature and use of the international system of units must be homogenized.